# A Cross-Cultural Evaluation of Liking and Perception of Salted Butter Produced from Different Feed Systems

**DOI:** 10.3390/foods9121767

**Published:** 2020-11-28

**Authors:** Emer C. Garvey, Thorsten Sander, Tom F. O’Callaghan, MaryAnne Drake, Shelley Fox, Maurice G. O’Sullivan, Joseph P. Kerry, Kieran N. Kilcawley

**Affiliations:** 1Food Quality & Sensory Department, Teagasc Food Research Centre, Moorepark, Fermoy, P61 C996 Co. Cork, Ireland; emer.garvey@teagasc.ie; 2Sensory Group, School of Food and Nutritional Science, University College Cork, T12 R220 Cork, Ireland; maurice.osullivan@ucc.ie; 3Department of Food, Nutrition, Facilities, FH Münster, Corrensstraße 25, D-48149 Münster, Germany; tsander@fh-muenster.de; 4Innovationsmanagement, Sensorische Produktevaluation und Consumer Trends, Marie-Jahn-Str.20, 30177 Hannover, Germany; 5School of Food and Nutritional Sciences, University College Cork, T12 Y337 Cork, Ireland; tom_ocallaghan@ucc.ie; 6Department of Food, Bioprocessing and Nutrition Sciences, Southeast Dairy Foods Research Center, North Carolina State University, Raleigh, NC 27695, USA; maryanne_drake@ncsu.edu; 7St. Angela’s Food Technology Centre, Lough Gill, 999928 Sligo, Ireland; sfox@stangelas.nuigalway.ie; 8Food Packaging Group, School of Food and Nutritional Science, University College Cork, T12 R220 Cork, Ireland; Joe.Kerry@ucc.ie

**Keywords:** dairy, diet, butter preference, sensory, volatiles

## Abstract

Perception and liking among Irish, German and USA consumers of salted butter produced from different feed systems—outdoor grass (FS-GRSS), grass/clover (FS-CLVR), and indoor concentrate (FS-TMR)—was investigated. A consumer study was conducted in all three countries. Irish and German assessors participated in ranking descriptive analysis (RDA), whereas descriptive analysis (DA) was carried out by a trained panel in the USA. Volatile analysis was conducted to identify differences in aroma compounds related to cow diet. Overall, there was no significant difference in overall liking of the butters, among USA, German and Irish consumers, although cross-cultural preferences were evident. Sensory attribute differences based on cow diet were evident across the three countries, as identified by German and Irish assessors and trained USA panelists, which are likely influenced by familiarity. The abundance of specific volatile aromatic compounds, especially some aldehydes and ketones, were significantly impacted by the feed system and may also contribute to some of the perceived sensory attribute differences in these butters.

## 1. Introduction

Globally, consumers are increasingly aware of their food choices, with respect to country of origin, production practices, sustainability, and potential health-promoting properties, prior to purchase [1]. Satisfaction of these extrinsic aspects can influence overall liking, and thus purchase intent and even willingness to pay a premium, particularly for meat and dairy products [2,3,4,5,6]. There has been substantial interest in exploring consumer’s perception towards meat and dairy products produced from a pasture/grazing diet. A recent review by Stampa, Schipmann-Schwarze et al. (2020) [7], mainly focusing on studies undertaken in Europe and the United States of America (USA), outlined that attitudes towards environmental practices and health benefits of consuming pasture-raised livestock products were the main drivers for consumption and willingness to pay premium prices for these types of products.

Farming practices in Ireland consist of fresh pasture for the majority of lactation, allowing for utilization of a low-cost, readily available feed source to produce high-quality milk products [8,9]. Since the abolishment of milk quotas in Europe in 2015, milk production in Ireland has increased substantially, with a 13% increase in milk intake from 2015 to 2017 [10,11,12]. In 2019, Ireland’s dairy sector grew in value by 11%, with butter being the largest export category [13] and further opportunities exist to expand current markets and develop new markets.

The most apparent differences in dairy products produced from cows on a pasture-rich diet, versus concentrates, are changes to the fatty acid (FA) profile and color. Inclusion of fresh pasture significantly increases levels of unsaturated FAs in milk [14,15,16], and β-carotene levels, enhancing a yellow color, particularly obvious in butter, as β-carotene is fat soluble [17]. Although milk and dairy products are not regarded as a dietary source of omega-3 FAs, higher amounts of α-linoleic acid and conjugated linoleic acid (CLA) are present in bovine milk from pasture based diets [14,15,18,19,20]. β-carotene derived from fresh pasture gives a yellow hue to butter, with the intensity of yellow positively correlating to the amount of fresh pasture grazed [21,22]. β-carotene is also a precursor for fat-soluble vitamin A, and a powerful antioxidant, and therefore also beneficial in the diet [23]. Butter hardness and spreadability are also dictated by the FA profile and thus impacted by the cow’s diet, with higher numbers of unsaturated FAs lowering the melting point [24].

Butter is coveted for its rich sensory attributes, with butter flavor being a significant driver for liking [25]. Although flavor is mainly dictated by the milk fat itself and added salt, volatile aroma compounds also play an important role in the sensory perception of butter. Previous studies have identified a range of potentially important volatiles including lactones, ketones, acids, esters, aldehydes, pyrroles and sulfur compounds, thought to influence the sensory perception of butter or sweet cream butter [26,27,28] and further work is required to determine factors that may influence the generation of these volatiles, such as cow diet. Although the influence of diet on the sensory properties of butter has been previously investigated [22,29], studies are limited and no studies have been published exploring the cross-cultural liking of salted butter. Therefore, an objective of this study was to investigate the liking and perception of salted butters, produced from cows outdoors on two pasture-based diets—perennial ryegrass, or perennial ryegrass/white clover—and cows indoors on a concentrate diet (total mixed rations) by consumers (Ireland, Germany and the USA), untrained assessors (Ireland and Germany) and trained panelists (USA). In addition, volatile analysis was performed to elucidate potential differences in sensory perception. The information generated should result in an improved understanding of the cross-cultural perception of Irish dairy products beneficial for the export markets.

## 2. Materials and Methods

### 2.1. Experimental Diets and Milk Production

Fifty-four spring calving Friesian cows were selected from the general herd at the Teagasc Animal and Grassland Research and Innovation Centre, Moorepark, Fermoy, Co. Cork. Teagasc has both an animal welfare body and animal ethics committee. The animal welfare body is a legal requirement of Article 26 of Directive 2010/63/EU and Regulation 50 of S.I. No. 543 of 2012. The Health Products Regulatory Authority provided project authorization, and the Health Products Regulatory Authority License number for this project is AE19132/P019. The cows were randomized based on milk yield, milk solids yield, calving date and lactation number, and allocated to one of three experimental feed systems (FS) (*n* = 18) (18 cows in each FS)—outdoor pasture grazing on perennial ryegrass (*Lolium perenne* L.) (FS-GRSS), outdoor pasture grazing on perennial ryegrass supplemented with white clover (*Trifolium repens* L.) (FS-CLVR) or housed indoor provided with a diet of TMR (FS-TMR). In-depth details of the three diets were outlined by O’Callaghan et al. (2016) [22]. Morning and evening milks from each of the experimental herds (FS-GRSS, FS-CLVR and FS-TMR) were collected and assigned as per feed system to 5000 L refrigerated tanks. Combined milk was kept at 4 °C prior to sample collection, which took place within 24 h after milking.

### 2.2. Butter Manufacture

Butter production took place on 3 separate occasions over a 3 week period. Butter produced from each experimental feed system was produced in triplicate (producing 3 batches per feed system). The procedure was identical to that as outlined by O’Callaghan, et al. (2016) [22].

The butter was packed into 200 ± 20 g sticks using an extruder and wrapped in grease-proof paper followed by an outer wrapping of aluminum foil. Butter was vacuum packed and stored at −20 °C until subsequent sensory and volatile analysis. Butter was defrosted in a refrigerator 24 h before the relevant analysis. Prior to each sensory study, the butter was tempered at room temperature for 1 h.

### 2.3. Consumer Study

#### 2.3.1. Consumer Selection

Three consumer sensory panels from three countries—Ireland, Germany and USA—were created for the purpose of this study. Panels were comprised of 108 German (70% female, 30 male, age = 18–68), 103 USA (79% female, 21% male, age = 21–64) and 96 Irish consumers (68% female, 32% male = 18–60), who regularly consumed butter. The German consumers consisted of a mixture of faculty and students, recruited in the University of Applied Sciences, Muenster, Germany. The USA consumers were recruited by the Southeast Dairy Foods Research Center, North Carolina State University, Raleigh, North Carolina, USA. The Irish consumers consisted of students and staff from Teagasc Food Research Centre, Moorepark, Fermoy, Co. Cork, Ireland and St. Angela’s Food Technology Centre, Sligo, Ireland.

#### 2.3.2. Product Evaluation by Hedonic, Intensity and Just-About-Right Scales

Butter evaluation took place in accordance with international standards [30], in each country. The ballots were identical in all three countries except that salt liking was not included in the USA ballot. The consumer study evaluation was designed to collect information on attribute liking, intensity perception and optimum levels of each attribute by applying a just-about-right scale (JAR). Liking was assessed for overall appearance, color, flavor, and texture attributes, intensity for color, flavor, saltiness, freshness, and firmness, and JAR scale evaluation was used to assess color, flavor, saltiness and texture. Each attribute was evaluated for liking and intensity on a 9-point hedonic scale; 9 = “extremely like” and 1 = “extremely dislike” and 9 = “high intensity” and 1 = “low intensity”, respectively (Appendix A). Five-point JAR scales were employed with extremes 1 = “much too little” and 5 = “much too much”. Since 9 butters were produced in total from the 3 experimental diets (3 butters produced in triplicate), the samples were presented to panelists in an incomplete block design, randomly allocated and single blinded using three-digit codes, allowing for even distribution of each treatment and butter trial. Therefore, the order of presentation was balanced, not randomized, and the ballot order for each product was identical as per standard consumer tests. Panelists were presented with three individual 5 ± 0.02 g servings of butter alongside water crackers and water. Crackers were provided for palatability purposes, the butter was presented on the crackers, but consumers were clearly instructed that the sensory evaluation was to be carried out solely on the sensory attributes of the butter. Panelists were encouraged to rinse their palate thoroughly after each sample and a 1 min rest between each sample was enforced. Consumers from the University of Applied Sciences, Muenster and Teagasc Food Research Centre completed the ballot questionnaire on paper. Consumers at North Carolina State University and St. Angela’s College undertook the questionaire via Compusense Cloud (Guelph, Canada) and FIZZ (Biosystems, Couternon, France), respectively.

### 2.4. Ranking Descriptive Analysis

Ranking descriptive analysis (RDA) [31], a modification of flash profiling, was performed by untrained assessors at the University of Applied Sciences, Muenster, Germany, and at the Teagasc Food Research Centre, Ireland. However, as stated, assessors were recruited on their frequency to consume butter, and had previous sensory analysis experience. Each panel was comprised of 20 German and 20 Irish assessors, respectively. Attributes were generated by a focus group consisting of 7 people, comprised of members from the Food Quality and Sensory Science Department at the Teagasc Food Research Centre, Ireland. The established list of attributes was chosen on their ability to best describe the different butter samples produced by each treatment. Irish and German assessors were asked to assess the attributes relative to color (yellow color), aroma (buttery, milky, grassy and rancid), flavor (salty, sweet, creamy, sour, stale) and texture (melt in the mouth) (Appendix A). Similar to the consumer study, each butter trial sample was presented to assessors on a water cracker for evaluation. The presentation for the RDA was a complete block design. Assessors were briefly coached on the explanation of each attribute in relevance to butter and asked to evaluate the intensity of each on a 9 cm continuous scale. Sensory analysis was conducted in duplicate over two separate occasions.

### 2.5. Descriptive Analysis Evaluation

Butter flavor was evaluated by a trained descriptive sensory panel using an established flavor language for butter [25] (Cooked/Nutty, Milkfat, Grassy, Mothball, Stale, Salty Taste and Color Intensity). All sensory testing was conducted in accordance with the North Carolina State University Institutional Review Board for Human Subjects guidelines. Panelists (*n* = 6) had more than 100 h of previous experience with the sensory analysis of dairy products. Prior to this study, panelists participated in 20 h of additional training on the three butters, FS-GRSS, FS-CLVR and FS-TMR to calibrate and confirm sensory attributes. Samples were prepared 24 h in advance and refrigerated at 4 °C. Prior to evaluation, butters were tempered to 15 °C. A cube of butter (~20 g) was placed in 3-digit-coded, 60 mL lidded cups (Sweetheart Cup Company, Owings Mills, MD, USA). Samples were evaluated on a 15-point intensity scale, in duplicate, on paper ballots by each panelist in a randomized balanced block design.

### 2.6. Volatile Analysis by HS-SPME-GC-MS

Volatile analysis was carried out by headspace solid-phase microextraction gas chromatography mass spectrometry (HS-SPME-GC-MS) utilizing a Gerstel MultiPurpose Sampler (GMPS) rail system (Anatune, Cambridge CB3 0NA, UK) connected to a Shimadzu GP2010 plus GC (Mason Technology Ltd., Dublin, Ireland). A 50/30 μm divinylbenzene/carboxen/polydimethylsiloxane (DVB/CAR/PDMS) SPME fiber was employed for analysis (Agilent Technologies Ireland Ltd., Cork, Ireland). The chosen HS-SPME parameters were as described by O’Callaghan [22], with modifications. More sample was used and a longer extraction time was applied in an attempt to recover a higher number compounds. Butter was thawed overnight at room temperature and 3 g was added to an La-Pha-Pack amber 20 mL screw-capped SPME vial with magnetic caps and silicone/polytetrafluoroethylene 1.3 mm 45° shore A septa (Apex Scientific Ltd., Co. Kildare, Ireland) and equilibrated for 10 min while exposed to a temperature of 40 °C, with pulsed agitation for 5 s at 350 rpm using the GMPS agitator/heater. The SPME fiber was exposed to the headspace above the samples, at a depth of 21 mm, for an extraction time of 60 min at 40 °C. The fiber was retracted, injected into the GC inlet with a merlin microseal (Merck, Arklow, Ireland) and desorbed for 3 min at 250 °C, followed by 3 min at 270 °C in the GMPS fiber bakeout station, to minimize carryover of compounds. Each butter trial was analyzed in triplicate. An external standard solution (1-butanol, dimethyl disulfide, butyl acetate, cyclohexanone) (Merck, Arklow, Ireland) at 1000 ppm in methanol (Merck, Arklow, Ireland) was also analyzed at the start and end of each batch, and levels of each external standard were quantified and compared to reference values to ensure that both the SPME extraction and MS detection were performing within specification.

The GC analysis was performed on a Shimadzu 2010 Plus GC (Mason Technology Ltd., Dublin, Ireland), equipped with a split/splitless injector, operating in the splitless mode. The carrier gas was helium held at a pressure of 43.8 psi and a flow rate of 1.2 mL/min. The volatile compounds were separated on a DB-624 UI (60 m × 0.32 mm × 1.80 μm) column (Agilent Technologies Ireland Ltd., Cork, Ireland). The temperature of the column oven was set at 40 °C, held for 5 min, increased at 5 °C/min to 230 °C then increased at 15 °C/min to 260 °C. The total GC run time was 65 min. Compound identification was carried out by a Shimadzu TQ8030 mass spectrometry detector (Mason Technologies Ltd., Dublin, Ireland) ran in the single-quad mode. The ion source temperature was 220 °C and the interface temperature was set at 260 °C. The MS mode was electronic ionization (70 eV) with the mass range scanned between *m*/*z* 35 and 250 amu. Compounds were identified using mass spectra comparisons to the NIST 2014 mass spectral library, the Shimadzu commercial library FFNSC (Flavor and Fragrance Natural and Synthetic Compounds library) version 2 and an in-house library created in GCMS Solutions software (Shimadzu, Japan) created with standards (where possible), and with target and qualifier ions and linear retention indices for each compound [32]. Spectral deconvolution was also performed to confirm identification of compounds using AMDIS (Automated Mass Spectral Deconvolution and Identification System, www.amdis.net).

### 2.7. Statistical Analysis

Data analysis was handled accordingly based on the normality of the data. Hedonic data from the sensory evaluation was analyzed using a non-parametric Kruskal–Wallis test (α = 0.05), with post hoc Mann-Whitney to identify the significant differences between samples. Bonferroni adjustment was applied to account for type 1 error, therefore working at an alpha level of 0.017. Analysis of variance (ANOVA) with post hoc Tukey significant test was applied to RDA and descriptive analysis data, working at an alpha level of 0.05. Just-about-right (JAR) data was assessed using chi-square statistic. A combination of parametric and non-parametric tests was used to evaluate the volatile data. All parametric and non-parametric tests were performed using SPSS IBM SPSS Statistics 24 for windows (SPSS Inc., IBM Corporation, NY, USA).

## 3. Results and Discussion

### 3.1. Consumer Evaluation

The average results of the sensory evaluation of consumers liking towards FS-GRSS, FS-CLVR and FS-TMR butters are presented in Table 1. Overall, there were no significant (*p* < 0.05) differences in overall liking of all three butters within each consumer cultural group; however, cross-cultural differences were evident.

#### 3.1.1. Irish Consumers

There was no significant (*p* > 0.05) difference among Irish consumers liking of the sensory attributes (overall appearance, color, flavor, saltiness, texture) or overall liking of the three butters (Hedonics Table 1). These results contradict a previous study by O’Callaghan et al. 2016 [22], who found that Irish consumers preferred the appearance and flavor of butters produced from grass and clover diets, compared to butter produced from TMR. However, in this study, Irish consumers did perceive the color intensity of FS-GRSS and FS-CLVR butters significantly (*p* < 0.017) higher than FS-TMR butter (Intensity Scale Evaluation Table 1), presumably due to higher β-carotene levels, with 41.6% of panelists grouping the FS-TMR butter as ‘not yellow enough’ (JAR Evaluation Table 1). However, this did not negatively influence their liking of the FS-TMR butter.

#### 3.1.2. German Consumers

Similar to Irish consumers, there was no significant (*p* > 0.05) difference in liking by German consumers for all three butters for sensory attributes, or for overall liking (Hedonics Table 1). In agreement with Irish consumers, German consumers rated the color of both FS-GRSS and FS-CLVR butter significantly (*p* < 0.017) more intense than FS-TMR butter (Intensity Scale Evaluation Table 1) and showed a significant (*p* < 0.05) higher score for ‘not yellow enough’ for FS-TMR butter (JAR Evaluation Table 1). German consumers also found the salt intensity of FS-GRSS and FS-CLVR butters, significantly (*p* < 0.017) higher than FS-TMR butter, even though identical levels of salt were added to each batch. Additionally, there was a significant (*p* < 0.017) difference in the perception of firmness intensity, with FS-TMR butter perceived as firmer than FS-GRSS and FS-CLVR butters. Although FA analysis was not undertaken in this study, butters from an identical experimental trial [22], and other studies [33,34], have characterized butter produced from pasture diets higher in unsaturated FAs, therefore the pasture-derived butter is likely to be softer, due to a lower melting point. The higher salty intensity perception of FS-GRSS and FS-CLVR butter as perceived by the German consumers may also relate to the softer texture of these butters and their behavior in the mouth—more rapid melting compared to FS-TMR butter (Figure 1). This would also appear to be confirmed by the JAR Evaluation of texture for FS TMR butter, which was deemed significantly (*p* < 0.05) higher for ‘much too firm’. Dadalı and Elmacı (2019) [35] found that margarine with the lowest score for hardness was perceived as the saltiest, despite having identical salt contents.

#### 3.1.3. USA Consumers

For overall appearance and color liking, USA consumers scored FS-CLVR and FS-TMR butters significantly (*p* < 0.017) higher than FS-GRSS butter (Hedonics Table 1). When attempting to elucidate the drivers for liking of butter, Krause et al. 2007 [25] identified from a focus group that USA consumers found a light yellow color desirable in butter, which likely explains why USA consumers in this study rated their liking of appearance and color of FS-TMR butter the highest (Hedonics Table 1), promoting the theory of familiarity dictating preference [36,37]. However, it is difficult to interpret why consumers liked the FS-CLVR butter similarly to FS-TMR butter for appearance and color, yet rated the FS-GRSS butter significantly lower for these same attributes. In the JAR Evaluation, USA consumers also scored FS-CLVR and FS-TMR butters significantly (*p* < 0.05) higher for ‘just about right’ for yellow color. However, these same consumers also scored FS-TMR butter significantly (*p* < 0.05) higher for ‘not enough yellow’, compared to FS-GRSS and FS-CLVR butter. Interestingly, FS-GRSS butter flavor was perceived as the most favorable by USA consumers. However, it was not significantly (*p* > 0.017) different from FS-TMR butter. This may reflect the trend in the USA for value added grass fed milk and other dairy products [38]. In terms of texture liking, USA consumers liked the FS-CLVR butter significantly (*p* < 0.017) more than FS-TMR butter. Krause et al. 2007 [25] also identified a cluster of USA consumers, referred to as ‘margarine lovers’, who were also butter users, but preferred the sensory attributes of margarines, i.e., the soft texture. This result corresponds to that of the JAR Evaluation, where consumers ranked FS-CLVR and FS-GRSS butters higher for ‘just-about-right’ texture, with FS-CLVR butter significantly (*p* < 0.05) higher compared to FS-TMR butter. In addition, USA consumers ranked FS-TMR butter significantly (*p* < 0.05) higher for ‘much too firm’.

### 3.2. Cross-Cultural Perceptions of Butters

Irish and USA consumers scored the overall appearance liking of FS-GRSS and FS-CLVR butters significantly (*p* < 0.017) higher compared to German consumers (Hedonics Table 1). However, Irish and German consumers both scored FS-TMR butter statistically (*p* < 0.017) lower for liking of appearance. Cross-cultural liking of color corresponds to liking of appearance, with Irish and USA consumers liking FS-CLVR butter significantly (*p* < 0.017) more than German consumers. Irish consumers are accustomed to yellow butter and previously showed highest liking for butter produced from grass and clover, compared to TMR [22]. Referring to the study by Krause et al. 2007 [25], the same cluster of USA consumers who liked softer ‘margarine like’ butters also preferred those butters and spreads which were darker in color, in agreement with this study. Similarly, USA consumers scored the FS-GRSS butter significantly (*p* < 0.017) higher for liking of flavor than Irish and German consumers. Both Irish and German consumers had similar liking for the texture of FS-GRSS butter, which was significantly (*p* < 0.017) higher compared to USA consumers. Overall acceptability of FS-GRSS butter was rated significantly (*p* < 0.017) higher by USA consumers compared to Irish and German consumers, which may be driven by their high liking for flavor. Both USA and Irish consumers considered FS-CLVR butter to be significantly (*p* < 0.017) higher for overall acceptability compared to German consumers. Salt intensity was perceived significantly (*p* < 0.017) higher by German consumers compared to Irish consumers for both FS-GRSS and FS-CLVR butter (Intensity Scale Evaluation Table 1). Irish consumers are familiar with soft butter, and did not perceive a significant difference in salt taste in agreement with a previous study [22]. Butter sold in Germany, however, is typically unsalted, which may explain the higher perceived salt intensity in the pasture butters, although texture, as discussed earlier, is also likely a contributory factor. Perception of freshness intensity was significantly lower (*p* < 0.017) by German consumers compared to Irish consumers for FS-CLVR butters. Irish consumers found FS-GRSS butter to be significantly (*p* < 0.017) firmer compared to German consumers, which again, is likely to be related to familiarity [22].

Significant (*p* < 0.017) differences were identified for liking of FS-TMR butter, with USA consumers rating the overall appearance, color and overall liking significantly (*p* < 0.017) higher, compared to Irish and German consumers, with a similar trend identified by Krause et al. 2007 [25]. There was no significant (*p* > 0.017) difference in liking of flavor of FS-TMR butter between Irish and USA consumers, which differed to O’ Callaghan et al. 2016 [22], where Irish consumers rated liking of flavor of TMR butter significantly lower than butter produced from milk from grass and clover diets. There was a significant (*p* < 0.017) difference in the liking of texture of FS-TMR butter by Irish and German consumers compared to USA consumers. For intensity ratings, Irish consumers ranked color, flavor and firmness of FS-TMR butter significantly (*p* < 0.05) higher than German consumers. No significant differences were evident in relation to JAR Evaluation for color, flavor, salt (USA consumers did not assess salt) or texture between the cultural groups. Overall, the cross-cultural perception of butter attributes by the three consumer groups was within a similar range on the hedonic scale, signifying a liking of butter among all three consumer groups.

### 3.3. Ranking Descriptive Analysis

The average results of Irish and German assessors perception of butters are portrayed in Figure 1a,b and in Table 2. Corresponding to the results for yellow intensity (Table 1), both Irish and German assessors rated FS-TMR butter significantly (*p* < 0.05) lower for yellow color. Both German and Irish assessors rated FS-CLVR butter as having a more intense yellow color (Figure 1a,b), compared to FS-GRSS butter, although not significant, this may suggest that FS-CLVR butter contained higher amounts of β-carotene (β-carotene levels vary with intake of grass and clover outdoors). O’Callaghan et al. 2016 [22] identified grass butters as having the highest levels of β-carotene. However, Panthi, Sundekilde et al. (2019) [39] identified Massdam cheeses produced from cows supplemented with white clover as having higher amounts of β-carotene compared to cheese produced from only grass. As mentioned, β-carotene content in milk will vary due to the levels of grass and clover ingested by cows due to differences in availability within the pasture. German assessors perceived FS-TMR butter as significantly (*p* < 0.05) darker than Irish assessors; however, this does not match results from the intensity scale portion of the consumer study (Table 1). This result may be due to the fact that in Ireland only butter derived from pasture is commercially available, while in Germany butter’s from both pasture or concentrate are widely available, and therefore Irish consumers may have scored TMR butter lower for color intensity due to a lack of familiarity.

Irish assessors perceived the aroma of FS-GRSS and FS-CLVR butters as significantly (*p* < 0.05) more buttery compared to FS-TMR butter, similar to O’Callaghan et al. 2016 [22], where consumers rated butter produced from grass significantly higher for diacetyl aroma compared to TMR butter. There was no significant difference (*p* < 0.05) detected among the German assessors for buttery aroma. German assessors perceived FS-GRSS butters to be significantly more milky compared to Irish assessors. Grassy and rancid aroma were not identified as significantly (*p* > 0.05) different by either German or Irish assessors. Both Irish and German assessors perceived significant (*p* < 0.05) differences in saltines perception, with values for FS-GRSS and FS-CLVR butters significantly higher, compared to FS-TMR butter. This is likely linked to the texture—melt in the mouth attribute—where Irish assessors perceived FS-GRSS and FS-CLVR butters to melt much more rapidly (*p* < 0.05) in the mouth. As previously mentioned, this is likely due to changes in fatty acid profile, which influence spreadability and melting properties of butter [40]. Similarly, Irish assessors perceived the flavor attribute creamy as significantly (*p* < 0.05) higher in both FS-GRSS and FS-CLVR butters, than in FS-TMR butter, which is also likely directly related to the texture, and in turn to the unsaturated fatty acid profile of these butters. Irish assessors have previously perceived pasture butters as significantly more creamy [22]. Irish assessors in this study also found that FS-GRSS and FS-CLVR butters were significantly higher (*p* < 0.05) for melt in the mouth than FS-TMR butter. German assessors scored melt in the mouth statistically higher (*p* < 0.05) for TS-TMR butter than Irish assessors. German assessors also found FS-CLVR butter significantly (*p* < 0.05) more sour compared to Irish assessors, and this may relate to the lower rating of freshness as perceived in the consumer study (Intensity Scale Evaluation Table 1). 

### 3.4. Descriptive Analysis Evaluation of FS-GRSS, FS-CLVR and FS-TMR Butters by Trained USA Panelists

The results of the descriptive analysis (DA) undertaken by trained USA panelists are listed in Table 3. There was no significant (*p* > 0.05) difference perceived for cooked/nutty, milkfat, and salty taste, between all three butters. Grassy was rated significantly (*p* < 0.05) higher in FS-GRSS butter, followed by FS-CLVR butter. However, it was not detected in FS-TMR butter, corresponding to results from Cheng et al. (2020) [41], who reported significant difference in grassy/hay perception of bovine skim milk powder, produced from pasture versus indoor TMR diets, as assessed by a descriptive panel. Villeneuve et al. 2013 [42] identified that grassy intensity was higher from milk produced by pasture and correlated to the higher levels of the aldehyde pentanal. The attribute stale was not perceived in any of the butters, and the levels of saltiness perception were similar (Table 3). Panelists found a significant (*p* < 0.05) difference in the color intensity of the three butters, in the following order: FS-CLVR > FS-GRSS > FS-TMR. Mothball flavor was noted in FS-CLVR butter but not in FS-GRSS or FS-TMR butters. This is a feed specific flavor that has been documented in previous studies (butter, cheese, dried ingredients) manufactured from pasture feeding, and appears to be specific to certain types of pastures [25,43].

### 3.5. Volatile Compounds

HS-SPME-GC-MS analysis of the butters identified a total of 30 volatile compounds across the three feeding systems (Table 4). Aldehydes, ketones, acids, terpenes and lactones were the main chemical classes contributing to the volatile profile of all three butters. We have only discussed compounds where the abundances are significantly different with respect to the feed systems.

The aldehyde compounds—pentanal, hexanal, heptanal and decanal—were most influenced by the different feed systems. All of these compounds are associated with lipid oxidation [44]. Levels of pentanal were significantly (*p* < 0.05) more abundant in FS-CLVR butter compared to FS-GRSS and FS-TMR butters in agreement with previous studies on butter [22], and pasteurized bovine milk [45]. Pentanal is derived from the fatty acids arachidonic and linoleic acid has the potential to adversely impact sensory perception by yielding a paint-like, cardboard aroma [45]. Hexanal, derived from linoleic acid [46], can confer a grassy off flavor in butter [47], and was significantly more abundant in FS-CLVR and FS-TMR butter, compared to FS-GRSS butter (Table 4). Heptanal was significantly (*p* < 0.017) more abundant in FS-CLVR and FS-GRSS butters in comparison to FS-TMR butter, and is also a product of linoleic acid, which has a green sweet aroma in dairy products [48]. Decanal, a compound of oleic acid degradation, was significantly (*p* < 0.05) more abundant in FS-CLVR butter compared to FS-GRSS butter and has been identified as having a green, fatty aroma in sweet cream butter [27]. Faulkner et al. 2018 [49] found a similar trend in decanal levels from raw milk produced from grass, grass/clover and TMR diets. Although the relative abundance of precursor unsaturated fatty acids is important, other factors such as the presence of natural pro- and anti-oxidants is also important.

Six ketones were identified in the three butter samples; 2,3-butanedione (diacetyl), a very odor-active compound with a characteristic buttery aroma [50,51], was significantly (*p* < 0.05) more abundant in FS-CLVR and FS-GRSS butters compared to FS-TMR butter (Table 4). Diacetyl was not detected in any of the butters by O’Callaghan et al. (2016) [22], but grass derived butter was rated higher for diacetyl aroma. It is difficult to discern why more diacetyl would be present in the butters produced from milk derived from pasture, but it could be that precursors of diacetyl are higher in those milks, due to different microbial activities in the rumen. Diacetyl is derived from pyruvate where α-acetolactate synthase converts it to α-acetolactate, which is subsequently converted to diacetyl by non-enzymatic oxidative decarboxylation [52]. FS-CLVR butter also had significantly (*p* < 0.05) more 2-butanone compared to FS-GRSS butter, with similar results seen in previous studies on butter [22] and milk [49]. This methyl ketone also derives from pyruvate metabolism [53], similarly to diacetyl. Acetone was significantly (*p* < 0.05) higher in FS-GRSS and FS-CLVR butters than in FS-TMR butter, and has also been identified as a product from concentrate feed [54]. Previous studies on milk and skim milk powder produced from pasture and TMR diets did not find differences in the abundance of acetone based on diet [40,49].

Both butanoic and nonanoic acid were significantly (*p* < 0.05) more abundant in FS-CLVR butter in comparison to FS-TMR and FS-GRSS butter. The presence and concentration of these acids are potentially important in flavor perception, with butanoic acid described as dirty sock in butter, and quite odor active [27]. Bovolenta et al. (2014) [55] found that when cows were exposed to high levels of pasture, there was a significant influence on nonanoic acid in Montasio cheese, with this compound not detected in cheese produced from milk derived from a low pasture diet.

Although not recognized to play such a significant role in odor and sensory quality of butter, toluene was significantly more abundant (*p* < 0.05) in both FS-CLVR and FS-GRSS butter than in FS-TMR butter. Toluene is a by-product of β-carotene metabolism [56], demonstrating its potential as a biomarker for pasture-derived dairy products [44]. As FS-CLVR butter was rated most yellow in this study, this may also suggest higher levels of β-carotene and hence the abundance of toluene.

The greater abundance of ethyl acetate in FS-CLVR butter is difficult to discern except that it is a product of ethanol and acetic acid. The sources of both precursors may be influenced by cow diet [45].

## 4. Conclusions

This research assessed the cross-cultural perception and liking of butters produced by three different feed systems. Overall, Irish, German, and USA consumers did not discriminate their overall liking of butters based on feed system, although some clear cultural differences were evident. Familiarity was postulated to contribute to differences in appearance liking and color liking by USA consumers, where the indoor TMR feed system scored highest. However, this did not impact overall liking of butter produced from pasture (FS-GRSS and FS-CLVR) by USA consumers. Both Irish and German consumers rated the color intensity of the pasture butters (FS-GRSS and FS-CLVR) higher than FS-TMR butter. German consumers also found that pasture (FS-GRSS and FS-CLVR) butter was more salty and that FS-TMR butter was firmer. It seems plausible that the texture and salty differences noted by German consumers are likely related to differences in their FA profile, which directly impacts on texture, but also indirectly on salty perception as the greater unsaturated FAs lower the melting point in the pasture butters. RDA by Irish and German assessors also identified important cross-cultural differences—German assessors perceived FS-GRSS butter as significantly more sour and milky, and scored FS-TMR butter higher for melt in the mouth, in comparison to Irish assessors. Both Irish and German assessors found that FS-TMR butter had less yellow color in agreement with the consumers, and German assessors found it darker than Irish assessors. Both Irish and German assessors found that FS-CLVR butter was the most intense yellow. USA panelists also found that color intensity ranged from FS-CLVR > FS-GRSS > FS-TMR. These results correlate with previous studies highlighting the impact of carotenoids, specifically β-carotene on the yellow color of pasture-derived dairy products. The trained USA panelists also found that a grassy flavor was highest in FS-GRSS butter but absent in FS-TMR butter, and that FS-CLV butter had a mothball attribute absent in FS-GRSS and FS-TMR butter. Volatile analysis identified a number of compounds that were statistically different based on diet—aldehydes, ketones, acids, toluene and ethyl acetate. The aldehydes, ketones and acids in the butter from the pasture diets may be influencing the grassy, milky and sour flavors, as perceived by USA and German consumers. Diacetyl may also be influencing enhanced buttery aroma, as perceived by Irish assessors in pasture (FS-GRSS and FS-CLVR) butter.

This study demonstrates that different feed systems affect cross-cultural perception of butter and that familiarity of products from specific feeds systems is a factor, but that it does not adversely impact butter acceptance in terms of overall liking within these cultural groups.

## Figures and Tables

**Figure 1 foods-09-01767-f001:**
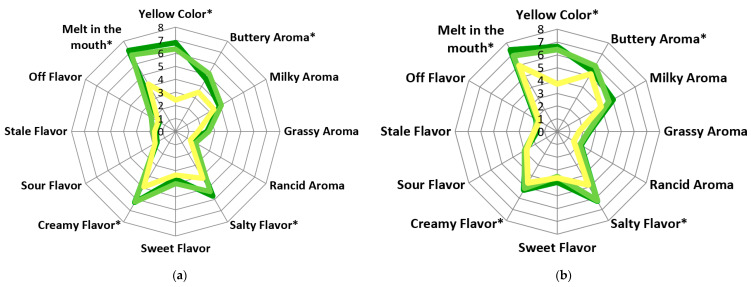
Average results (*n* = 20) from ranking descriptive analysis evaluation of FS-GRSS, FS-CLVR and FS-TMR butters by Irish (**a**) and German (**b**) assessors. * *p* < 0.05.

**Table 1 foods-09-01767-t001:** Cross-cultural comparison of liking, intensity rating and just-about-right scale evaluation by Irish, German and United States of America consumers, of butters produced by different diets—FS-GRSS, FS-CLVR and FS-TMR.

		Irish Consumers			German Consumers			USA Consumers		
		FS-CLVR	FS-GRSS	FS-TMR	FS-CLVR	FS-GRSS	FS-TMR	FS-CLVR	FS-GRSS	FS-TMR
Hedonics										
Overall appearance		6.51 ± 1.56 ^x^	6.46 ± 1.65 ^x^	5.99 ± 1.76 ^y^	5.62 ± 1.59 ^y^	5.69 ± 1.65 ^y^	5.44 ± 1.69 ^y^	6.46 ± 1.63 ^abx^	6.33 ± 1.57 ^bx^	6.98 ± 1.42 ^ax^
Color		6.44 ± 1.61 ^x^	6.19 ± 1.80	5.80 ± 1.80 ^y^	5.66 ± 1.50 ^y^	5.88 ± 1.56	5.56 ± 1.38 ^y^	6.43 ± 1.64 ^abx^	6.18 ± 1.72 ^b^	6.91 ± 1.57 ^ax^
Flavor		6.46 ± 1.74	6.44 ± 1.67 ^y^	6.28 ± 1.68 ^x^	6.19 ± 1.65	6.38 ± 1.66 ^y^	5.89 ± 1.69 ^y^	6.52 ± 1.75 ^b^	7.10 ± 1.55 ^ax^	6.76 ± 1.79 ^abx^
Salt		6.00 ± 1.63	5.73 ± 1.79	5.78 ± 1.68	5.83 ± 0.75	5.87 ± 0.73	5.52 ± 0.72			
Texture (firmness for USA)		6.54 ± 1.58	6.35 ± 1.72 ^x^	6.05 ± 1.75 ^x^	6.40 ± 1.84	6.52 ± 1.51 ^x^	6.17 ± 1.84 ^x^	6.26 ± 1.67 ^a^	5.95 ± 1.69 ^aby^	5.50 ± 1.81 ^by^
Overall liking		6.46 ± 1.67 ^x^	6.36 ± 1.76 ^y^	6.27 ± 1.66 ^y^	6.04 ± 1.74 ^y^	6.23 ± 1.61 ^y^	5.89 ± 1.90 ^z^	6.59 ± 1.65 ^x^	7.13 ± 1.47 ^xy^	6.85 ± 1.67 ^x^
Intensity Scale Evaluation										
Color		6.14 ± 1.89 ^a^	5.68 ± 2.41 ^ab^	5.22 ± 2.40 ^bx^	6.31 ± 0.80 ^a^	6.02 ± 0.76 ^a^	3.27 ± 0.81 ^ay^			
Flavor		6.34 ± 1.78	5.73 ± 2.01	6.21 ± 1.60 ^x^	6.38 ± 1.79 ^a^	6.14 ± 1.87 ^a^	5.29 ± 2.23 ^by^			
Salt		5.31 ± 2.00 ^y^	5.15 ± 2.15 ^y^	5.06 ± 1.96	6.17 ± 1.88 ^ax^	5.95 ± 1.81 ^ax^	5.28 ± 1.89 ^b^			
Freshness		6.34 ± 1.73 ^x^	6.10 ± 1.93	5.95 ± 1.82	5.56 ± 0.89 ^y^	5.78 ± 0.82	5.80 ± 0.97			
Firmness		5.46 ± 2.05 ^x^	5.34 ± 2.15 ^x^	5.61 ± 1.94 ^x^	3.62 ± 2.08 ^by^	3.42 ± 2.06 ^by^	4.53 ± 1.98 ^ay^			
JAR Evaluation										
Color	Not Yellow Enough	23.96%	31.25%	41.67%	7.41% ^b^	9.26% ^b^	51.85% ^a^	2.9% ^b^	4.9% ^b^	26.2% ^a^
	Just about Right	62.50%	53.125%	42.71%	37.96%	38.89%	42.59%	57.3% ^ab^	49.5% ^b^	72.8% ^a^
	Too Yellow	13.54%	15.625%	15.63%	54.63%	51.85%	5.56%	39.8%	45.6%	1.0%
Flavor	Not Enough Flavor	18.75%	26.04%	17.71%	18.52%	24.07%	42.59%	26.2%	21.4%	33.0%
	Just about Right	62.50%	60.42%	56.25%	51.85%	53.70%	41.67%	58.3%	70.9%	63.1%
	Too Much Flavor	18.75%	13.54%	26.04%	29.63%	22.22%	15.74%	15.5% ^a^	7.8% ^ab^	3.9% ^b^
Salt	Not Enough Salt	18.75%	27.08%	16.67%	17.59%	21.30%	30.56%			
	Just about Right	62.50%	51.04%	59.38%	38.89%	41.67%	37.04%			
	Too Much Salt	18.75%	21.88%	23.96%	43.52%	34.26%	32.41%			
Texture	Not Firm Enough	2.08%	10.42%	10.42%	42.59%	38.89%	19.44%	0.0%	0.0%	1.0%
	Just about Right	76.04	63.54	52.08	55.56%	57.41%	64.81%	58.3% ^a^	48.5% ^ab^	35.9% ^b^
	Much Too Firm	21.88	26.04	37.50	1.85%^b^	3.70% ^b^	15.74% ^a^	41.7% ^b^	51.5% ^ab^	63.1% ^a^

Within each consumer cultural group: values in the same row not sharing the same superscript (a, b) indicate significant difference (identified using Kruskal–Wallis and Mann–Whitney test for multiple comparison, α = 0.017). Cross-cultural comparison: values in the same row not sharing the same superscript (x, y, z) indicate significant difference (identified using Kruskal–Wallis and Mann–Whitney test for multiple comparison, α = 0.017). Values provided after ± are standard deviations.

**Table 2 foods-09-01767-t002:** Cross-cultural comparison of RDA evaluation by Irish and German assessors of butters produced by different diets—FS-GRSS, FS-CLVR and FS-TMR.

	Irish Assessors			German Assessors		
	FS-GRSS	FS-CLVR	FS-TMR	FS-GRSS	FS-CLVR	FS-TMR
Color						
Yellow color	6.81 ± 0.9 ^a^	6.33 ± 1.37 ^a^	2.41 ± 0.85 ^by^	6.63 ± 1.12 ^a^	6.41 ± 1.46 ^a^	3.72 ± 1.37 ^bx^
Aroma						
Buttery	4.66 ± 1.53 ^a^	5.13 ± 1.51 ^a^	3.48 ± 1.22 ^b^	5.48 ± 1.86	5.93 ± 1.99	5.20 ± 1.93
Milky	3.89 ± 1.45 ^y^	4.05 ± 1.29	3.40 ± 1.24	5.03 ± 2.56 ^x^	4.66 ± 2.25	3.93 ± 2.08
Grassy	2.41 ± 1.17	2.54 ± 1.34	1.71 ± 0.72	2.50 ± 1.92	2.42 ± 1.97	1.69 ± 1.11
Rancid	1.39 ± 0.69	1.76 ± 1.40	1.31 ± 0.73	2.11 ± 1.76	2.05 ± 1.64	1.49 ± 1.08
Flavor						
Salty	5.71 ± 1.13 ^a^	5.33 ± 1.66 ^a^	4.16 ± 1.77 ^b^	6.24 ± 2.00 ^a^	6.23 ± 1.47 ^a^	4.81 ± 1.77 ^b^
Sweet	3.64 ± 1.50	3.99 ± 1.45	3.34 ± 1.41	3.92 ± 1.95	3.50 ± 1.76	3.61 ± 1.84
Creamy	6.29 ± 1.25 ^a^	6.24 ± 1.37 ^a^	4.91 ± 1.46 ^b^	5.27 ± 1.90	5.10 ± 1.90	4.60 ± 2.03
Sour	1.64 ± 0.57 ^y^	1.76 ± 1.15	1.83 ± 0.79	2.85 ± 1.67 ^x^	2.97 ± 2.20	2.79 ± 1.82
Stale	1.51 ± 0.81	1.76 ± 1.07	1.61 ± 0.74	1.59 ± 0.78	1.80 ± 1.49	1.83 ± 1.44
Off flavor	1.44 ± 0.63	2.16 ± 1.84	1.56 ± 1.01	1.90 ± 1.33	1.81 ± 1.25	1.66 ± 1.33
Texture						
Melt in the mouth	7.19 ± 0.85 ^a^	6.79 ± 1.26 ^a^	4.19 ± 1.73 ^by^	7.36 ± 1.20	6.86 ± 1.46	5.96 ± 2.21 ^x^

Values are the average results ±standard deviations. Within consumer group: values in the same row not sharing the same superscript (a, b) indicate significant difference (Confidence level 5%, identified using ANOVA and Tukey post). Cross-cultural comparison: values in the same row not sharing the same superscript (x, y) indicate significant difference (confidence level 5%, identified using student *t*-test).

**Table 3 foods-09-01767-t003:** Sensory attribute means from trained USA panel evaluation of FS-GRSS, FS-CLVR and FS-TMR butters. Table presents DA averages and standard deviations.

	Feed System		
Sensory Attribute	FS-CLVR	FS-GRSS	FS-TMR
Cooked/Nutty	3.1 ± 0.1	3.06 ± 0.2	3.3 ± 0.1
Milkfat	3.1 ± 0.1	3.1 ± 0.2	3.2 ± 0.1
Grassy	1.2 ± 0.1 ^b^	1.4 ± 0.1 ^a^	ND ^c^
Mothball	1.3 ± 0.1 ^a^	ND ^b^	ND ^b^
Stale	ND	ND	ND
Salty Taste	11.1 ± 0.1	10.9 ± 0.7	11.2 ± 0.1
Color Intensity	4.2 ± 0.1 ^a^	3.4 ± 0.3 ^b^	1.8 ± 0.1 ^c^

Means represent duplicate evaluations from three experimental replications by six highly trained panelists. Attributes were scored using a 0 to 15 point universal intensity scale consistent with the Spectrum descriptive analysis method. ND—not detected. Means in a row followed by different superscript letters are different (*p* < 0.05). Values provided after ± are standard deviations.

**Table 4 foods-09-01767-t004:** Average (*n* = 9) peak area values (×10^6^) of volatile compounds identified in FS-GRSS, FS-CLVR and FS-TMR butters.

					Feeding System		
Volatile Compound	CAS NUMBER ^#^	Odor Descriptors	RI	REF RI	FS-GRSS	FS-CLVR	FS-TMR
Aldehyde							
Pentanal ^2^	110-62-3	Pungent, almond like, chemical, malty, apple [22]	731	733	0.054 ± 0.027 ^b^	0.489 ± 0.394 ^a^	0.049 ± 0.018 ^b^
Hexanal ^3^	66-25-1	Green, slightly fruity, lemon, herbal, grassy [22]	836	839	0.032 ± 0.013 ^b^	0.060 ± 0.035 ^a^	0.049 ± 0.018 ^a^
Heptanal ^3^	111-71-7	Slightly fruity (balsam), fatty, oily [22]	937	943	0.020 ± 0.008 ^a^	0.035± 0.027 ^a^	0.010 ± 0.004 ^b^
Benzaldehyde	100-52-7	Bitter, almond, sweet cherry [22]	1026	1028.9	0.020 ± 0.012	0.019 ± 0.011	0.028 ± 0.011
Nonanal	124-19-6	Green, citrus, fatty, floral [22]	1143	1150	0.039 ± 0.040	0.035 ± 0.028	0.024 ± 0.017
Decanal ^3^	112-31-2	Green [28]	1246	-	0.001 ± 0.001 ^b^	0.003± 0.001 ^a^	0.002 ± 0.002 ^ab^
Ketone							
Acetone^2^	67-64-1	Earthy, strong fruity, wood pulp, hay [22]	529	533	0.224 ± 0.125 ^ab^	0.216 ± 0.052 ^a^	0.124 ± 0.027 ^b^
Diacetyl (2,3-Butanedione) ^2^	431-03-8	Buttery [50]	628	-	0.044 ± 0.015 ^a^	0.079 ± 0.057 ^ab^	0.022 ± 0.009 ^b^
2-Butanone ^2^	78-93-3	Buttery, sour milk, etheric [22]	635	639	0.106 ± 0.019 ^b^	0.286 ± 0.156 ^a^	0.170 ± 0.082 ^ab^
2-Heptanone	113-43-0	Blue cheese, spicy, Roquefort [22]	929	936	0.062 ± 0.011	0.077 ± 0.034	0.060 ± 0.007
2-Nonanone	821-55-6	Malty, fruity, hot milk, smoked cheese [22]	1133	1140	0.017 ± 0.003	0.037 ± 0.033	0.013 ± 0.005
2-Pentanone	107-87-9	Orange peel, sweet, Fruity [22]	725	-	0.029 ± 0.012	0.032 ± 0.010	0.031 ± 0.014
Acid							
Butanoic Acid ^1^	107-92-6	Sweaty, butter, cheese, Strong, acid, fecal, rancid [22]	860	864	0.015 ± 0.007 ^b^	0.024 ± 0.004 ^a^	0.017 ± 0.008 ^ab^
Hexanoic Acid ^1^	142-62-1	Acidic, sweaty, cheesy, sharp, goaty, bad breath [22]	1045	1052	0.023 ± 0.009	0.041 ± 0.02	0.026 ± 0.010
Nonanoic Acid ^2^	112-05-0	Waxy, dirty and cheesy with a cultured dairy nuance **	22.7	-	0.014 ± 0.007 ^b^	0.026 ± 0.005 ^a^	0.014 ± 0.008 ^b^
Hydrocarbons							
Toluene^2^	108-88-3	Nutty, bitter, almond, Plastic [22]	789	794	0.794 ± 0.26 ^b^	1.793 ± 0.708 ^a^	0.139 ± 0.031 ^c^
* o-Xylene	108-38-3	Geranium **	895	-	0.371 ± 0.304	0.303 ± 0.386	0.572 ± 0.449
* *p*-Xylene	106-42-3	Not listed **	923	-	0.177 ± 0.187	0.087 ± 0.143	0.222 ± 0.136
Lactone							
δ-Hexalactone	823-22-3	Creamy, chocolate, sweet aromatic [50]	1215	-	0.129 ± 0.033	0.114 ± 0.026	0.117± 0.022
δ-Octalactone	698-76-0	Coconut like, peach [50]	1413	-	0.026 ± 0.006	0.024 ± 0.007	0.022 ± 0.005
δ-Decalactone	705-86-2	Coconut like, peach [50]	1691	1620.9	0.020 ± 0.007	0.021 ± 0.010	0.019 ± 0.008
Sulfide							
Dimethyl Sulfide	75-18-3	Corn like, fresh pumpkin [50]	534	538	0.011 ± 0.010	0.018 ± 0.016	0.009 ± 0.002
Carbon Disulfide	75-15-0	Sulfury, onion, sweet corn, vegetable, cabbage, tomato, green, radish **	542	548	0.057 ± 0.018	0.132 ± 0.115	0.088 ± 0.088
Ester							
Ethyl Acetate ^2^	141-78-6	Solvent, pineapple, Fruity, fruit gum [22]	639	-	0.013 ± 0.010 ^ab^	0.033 ± 0.019 ^a^	0.010 ± 0.009 ^b^
Ethyl Benzene	100-41-4	Not listed **	887	-	0.072 ± 0.046	0.103 ± 0.160	0.206 ± 0.213
Diethyl Ether	60-29-7	Ethereal **	512	-	0.020 ± 0.022	0.024 ± 0.025	0.024 ± 0.026
Other							
Ethanol	64-17-5	Alcoholic, ethereal, medicinal **	503	506	0.038 ± 0.020	0.032 ± 0.018	0.028 ± 0.021
1-Pentene	109-67-1	Not listed **	565	-	0.061 ± 0.024	0.047 ± 0.024	0.076 ± 0.077
α-Pinene	80-56-8	Mint, pine oil [27]	950	951	0.038 ± 0.032	0.014 ± 0.010	0.036 ± 0.026
Dodecane	112-40-3	Alkane **	1193	-	0.006 ± 0.004	0.007 ± 0.002	0.005 ± 0.002

RI: Retention index. REF RI: Reference retention index. ^#^ CAS: Chemical Abstracts Service Number. Values in the same row not sharing the same superscript (a, b) specify significant difference in peak area value average. Volatile compound annotated with a ^1^ indicate statistical analysis carried out by ANOVA and Tukey post hoc test, α = 0.05. Volatile compounds annotated with a ^2^ indicate statistical analysis carried out by Welch test and Games-Howell post hoc test, α = 0.05. Volatile compounds annotated with a ^3^ indicate statistical analysis carried out by Kruskal–Wallis and Mann–Whitney, α = 0.017. * Tentative identification due to isomers. Values provided after ± are standard deviations. ** Odor descriptors sourced from http://www.thegoodscentscompany.com/.- No REF RI available.

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
