# Peer review of "A Cross-Cultural Evaluation of Liking and Perception of Salted Butter Produced from Different Feed Systems"

_foods, 2020, doi:10.3390/foods9121767_

Round 1
Reviewer 1 Report
Foods (ISSN 2304-8158)
Manuscript ID: foods-1007116
Title: A cross-cultural evaluation of liking and perception of salted butter produced from different feed systems.
This manuscript investigates the liking and perception of salted butter produced from different feed systems in consumers from different countries. The topic is interesting and the results enrich the literature on mountain dairy products. The paper is written with care and results are clearly explained.
Nevertheless, important information are missing in Materials and Methods chapter so I think the paper needs a review.
line 65-70 in my opinion, here authors need to clarify why they proposed this study. What is missing from the existing literature?
line 71-75 Here is missing, in my opinion, a description of the research objectives, that is, the questions the authors have asked themselves. This paragraph, as it stands, basically describes what the authors did, are the answers to the questions.
line 81 it is not clear in my opinion if n = 18 is the total of the three experiments or if there are 18 cows for each feed. Please clarify.
line 94 is missing a description of how then the samples were prepared for sensory analysis, texture and volatile analysis. Were they made on the frozen product? At room temperature? Which? In what times? How soon before the analysis?
line 97 I state that I am not a native speaker and maybe I am wrong but I think that 'demographics' is not the right term here. It is true that different groups from different countries will probably have different demographic characteristics, but I think it would be more correct to put 'different countries' or 'different cultural groups'. Furthermore, demographic data are actually not reported here, please report for each group the % of females, the age range, the % of students ....
line 109 were the three tasks that consumers had to perform (liking, intensity and JAR) always proposed in this order or were they randomized? If the presentation was the same for everyone, it is important to explain this choice. Was a new sample set provided for each task to avoid the halo effect?
lines 118-121 it is not clear to me whether the tasting instruction included crackers or not. Was the consumer left free to choose whether to taste the butter alone or with the crackers? If so, even if the indication was to consider only the characteristics of the butter, the data are biased because the conditions are not the same and worse they are not controlled: it is not known who evaluated the butter alone and who with crackers.
line 128 a very brief description of the method and a reference should be provided here. The method is only about ten years old and is not that popular.
line 134 even if the data have been reported in table S1, I think that the text should be integrated by reporting the descriptors, as done in the results on lines 297-299.
line 137 for RDA, it is missing a description of sample preparation and how samples were served.
line 137-140 I believe the verb is missing
line 140 table S3 is not in the supplementary materials
line 148 the description of sample presentation is missing
line 194 the title is too long, in my opinion, I would keep 'consumer evaluation'
line 198 I would change 'demographics' with cultural
line 212 replace re with for
line 256 here too, the title is too long I would keep 'Cross-cultural perceptions'
line 294 I would replace demographics with 'consumer groups'
line 295 keep just ‘Ranking descriptive analysis'
line 297-299 I would move this information to materials and methods chapter in paragraph 2.4
line 311 how could this result be justified?
table 3 given the high number of tables, perhaps table 3 could easily be converted into a multiple bar chart with the same information.
All tables report averages with an error, what is it?
Report in the header if it is standard error or standard deviation.
Author Response
Please see the attached file where we have addressed each of your queries.
Thank you for your time.

Reviewer 2 Report
The manuscript shows a detailed study about consumer preference of salted butter produced from three feed system (outdoor grass, grass/clover and indoor). The study was performed in three different countries (Ireland, USA and Germany) using a ranking descriptive analysis and a trained panel. Results are coherent with objectives of the study. In general, overall liking of salted butters are not impacted by different culture.
Line 75. Remove “volatile organic compound” and write just “volatile compound”
Line 397. The section discuss about volatile compounds that have a chromatographic area is significative different depending of feed system. However, this article has a sensory view. Then, it is missing a discussion about even if the chromatographic area is not significative different a small variation in the concentration of the compound may have an impact on aroma perception. Moreover, the odour threshold is something interesting to discuss.
Table 4 must also include the odour described in the bibliography. Example: Toluene “glue, paint, solvent”; Pentanal “almond, green, malt, oil, pungent”; Hexanal “fresh, fruit, grass, green, oil”, etc.
Author Response
Please see attached file that has our responses to your queries.
Thank you for your time.

Reviewer 3 Report
The manuscript of "A cross-cultural evaluation of liking and perception of salted butter produced from different feed systems" presented an interesting topic related to a sensory/consumer analysis conducted between three different countries.
The sensory study is very well described, presented and discussed. In the case of consumer study, in all countries, the minimum of 100 participants should be preserved.
However, I cannot agree with the statement: "The abundance of specific volatile aromatic compounds, especially some aldehydes and ketones were significantly impacted by the feed system and are likely responsible for some of the perceived sensory attribute difference in these butters". In my opinion, Authors should be more specific here. PLS method can be used to study correlations between sensory attributes and volatile presence.
In the volatile analysis, did Authors selected the parameters of SPME by themselves or refers to previous research. (chose of fiber, temperature and time of extraction). What references were used for REF RI?
Some minor things:
In the title, there should not be a dot.
Keywords shouldn't be the same as the title: cross-cultural, butter, feed systems.
In the abstract, three times appeared the phrase: "feed system", it is worthy to re-change this.
Author Response
Please see our response to your queries in the attached file.
Thank you for your time.

Round 2
Reviewer 1 Report
The authors have made an effort to meet the demands and adjust the text.